# Identification of a Putative CFSH Receptor Inhibiting IAG Expression in Crabs

**DOI:** 10.3390/ijms241512240

**Published:** 2023-07-31

**Authors:** Fang Liu, Lin Huang, An Liu, Qingling Jiang, Huiyang Huang, Haihui Ye

**Affiliations:** 1Fisheries College, Jimei University, Xiamen 361021, China; liufang@jmu.edu.cn (F.L.); liuan@jmu.edu.cn (A.L.); 2College of Ocean and Earth Sciences, Xiamen University, Xiamen 361102, China; huanglin@stu.xmu.edu.cn (L.H.); jangqingling@163.com (Q.J.); huiyang@xmu.edu.cn (H.H.)

**Keywords:** CFSH, CFSHR, receptor identification, SEFIR, IAG, crustaceans

## Abstract

The crustacean female sex hormone (CFSH) is a neurohormone peculiar to crustaceans that plays a vital role in sexual differentiation. This includes the preservation and establishment of secondary female sexual traits, as well as the inhibition of insulin-like androgenic gland factor (IAG) expression in the androgenic gland (AG). There have been no reports of CFSH receptors in crustaceans up to this point. In this study, we identified a candidate CFSH receptor from the mud crab *Scylla paramamosain* (named *Sp*-SEFIR) via protein interaction experiments and biological function experiments. Results of GST pull-down assays indicated that *Sp*-SEFIR could combine with *Sp-*CFSH. Findings of in vitro and in vivo interference investigations exhibited that knockdown of *Sp*-SEFIR could significantly induce *Sp*-IAG and *Sp*-STAT expression in the AG. In brief, *Sp*-SEFIR is a potential CFSH receptor in *S. paramamosain*, and *Sp*-CFSH controls *Sp*-IAG production through the CFSH-SEFIR-STAT-IAG axis.

## 1. Introduction

In dioecious crustaceans, there are apparent differences between males and females, and this sexual dimorphism is caused by sex determination and differentiation [1,2]. At present, studies on the regulatory mechanisms of sexual differentiation mainly focus on endocrine regulation and related transcription factors [1,3,4,5]. Crustaceans have unique endocrine systems. Insulin-like androgenic gland (IAG) hormone generated from the androgenic gland (AG) and crustacean female sex hormone (CFSH) secreted by the eyestalk ganglion are considered vital hormones that regulate sexual differentiation [6,7]. It is widely recognized that IAG acts principally in male sexual differentiation [6]. Although IAG receptors have been found in a variety of species [8,9,10,11,12,13], no CFSH receptor (CFSHR) has been identified.

In 2014, the identification of CFSH was initially reported from the eyestalk ganglion of the Atlantic blue crab *Callinectes sapidus* [7]. Subsequently, its central functions in female sexual differentiation have been well-established in a number of species [7,14,15,16]. Aside from the eyestalk ganglion, CFSH transcripts have been detected in various tissues of many species, suggesting that CFSH has multiple biological functions [17,18,19,20,21,22,23,24,25,26]. The potential relationship between CFSH and female reproduction was speculated in the kuruma prawn *Marsupenaeus japonicus* and the giant freshwater prawn *Macrobrachium rosenbergii* [22,24]. Of note, results of recent investigations estimated that CFSH could also regulate differentiating the sex of males through inhibiting *IAG* expression in AG [15,16,27]. In the mud crab *Scylla paramamosain* as well as the Chinese mitten crab *Eriocheir sinensis*, transcripts of CFSH were detected in both males and females. Moreover, CFSH may suppress *IAG* expression in AG of these two species [16,27]. Moreover, identical outcomes were reported in hermaphroditic shrimp *Lysmata vittata* [15]. In the Chinese mitten crab *E. sinensis*, silencing of *CFSH1* expression significantly stimulated the development of the male external reproductive appendage [16]. A previous research study has demonstrated that CFSH has the ability to inhibit the expression of *signal transmitters and activators of transcription* (STAT), which results in *IAG* expression suppression in the mud crab *S. paramamosain* [28]. These results suggested that AG was the principal target organ of CFSH in males. Moreover, CFSH receptors regarding male sexual differentiation are likely to be identified from AG.

All identified CFSHs have two conserved domains: Cys-knot core structure and interleukin-17 (IL-17) domain [29,30]. It is speculated that CFSH evolves from an original protein that is similar to IL-17 [24]. According to the results of multiple sequence alignment and phylogenetic analysis, it can be inferred that CFSH exhibits homology to IL-17 and displays a high degree of conservation in Brachyura [27]. It provides ideas for predicting the possible receptor of CFSH. In 2006, the genome of the purple sea urchin *Strongylocentrotus purpuratus* proved the presence of IL-17 and interleukin-17 receptor (IL-17R), marking the initial documentation of IL-17 and associated signaling pathways in invertebrates [31,32]. Subsequently, IL-17 and its associated signaling molecules have been documented in numerous invertebrate species [33,34,35,36,37,38]. Although IL-17 showed low sequence homology among invertebrates, it has very conservative amino acid sites, especially four cysteine sites, which play a crucial role in the maintenance of its three-dimensional structure [39]. According to prior research studies, IL-17 controls the downstream gene expression via binding to its receptor (IL-17R) and activating STAT via the Janus kinases/signal transducers and stimulators of transcription (JAK/STAT) signaling pathway [40]. While it is important to note that structural resemblance does not always indicate functional resemblance, the IL-17 signaling pathway presents an exciting prospect for investigating the prediction of CFSHR. Therefore, we hypothesized that CFSHR was a SEFIR (to have a similar expression as the fibroblast growth factor (SEF) and IL-17Rs) domain that contained receptors similar to IL-17R.

In most commercial crustacean species, biological and economic traits differ between males and females. Large-scale mono-sex culture through sex control technology can improve production and economic benefits. *S. paramamosain* is a major aquaculture species in many countries of the Indo-West Pacific region. The regulatory role of CFSH in the process of female sexual differentiation of this particular species has been demonstrated [14]. The identification of CFSH receptors will provide important theoretical support for elaboration of the mechanism of sexual differentiation of *S. paramamosain* and the mono-sex culture of mud crabs. Recently, a protein containing the SEFIR domain was identified in the AG transcriptome library of *S. paramamosain*. Herein, we cloned the cDNA of *Sp*-SEFIR and explored its expression profiles. Subsequently, we identified *Sp*-SEFIR as the receptor for CFSH through protein interaction experiments and biological function experiments.

## 2. Results

### 2.1. Molecular Cloning and Phylogenetics of Sp-SEFIR

The full length of *Sp*-SEFIR (GenBank accession number: ON787957) cDNA is 2515 base pairs (bp), including a 70 bp 5′ untranslated region (UTR), a 1959 bp ORF and a 486 bp 3′ UTR. The polyadenylation signal (AATAAA) is located at 149 bp upstream of the polyA sequence (Figure A1). The ORF segment was responsible for the synthesis of a polypeptide consisting of 652 amino acids (aa), with a molecular weight of 73801.91 Da and a theoretical isoelectric point (pI) of 5.27. The precursor polypeptide was inferred to possess a 23 aa signal peptide, a 23 aa transmembrane domain, a 123 aa SEFIR domain and two low-compositional complexity regions (11 and 30 aa, respectively) (Figure 1A).

Phylogenetic analysis demonstrated that IL-17R was clustered into five clusters: IL-17RA, IL-17RB, IL-17RC, IL-17RD, as well as IL-17RE. *Sp*-SEFIR was classified as IL-17RD (Figure 1B).

### 2.2. Tissue Distribution of Sp-SEFIR

RT-PCR demonstrated that *Sp*-SEFIR had a broad distribution across different tissues in males. The *Sp*-SEFIR mRNA expression level was relatively higher in the Y organ, testis, androgenic gland, stomach, hepatopancreas and muscle (Figure 2A).

### 2.3. Expression Profile of Sp-SEFIR during AG Development

The expression profile of *Sp*-SEFIR in AG throughout AG advancement (stage I–III) was determined using qRT-PCR. The findings revealed that the expression patterns of *Sp*-SEFIR mRNA were elevated along with the development of AG to reach a peak at stage II before significantly being reduced at stage III (Figure 2B).

### 2.4. Immunofluorescence Localization of Sp-SEFIR in AG

Consistent with the previous study, two forms of glandular cells (A and B) were observed in AG [41]. Type A glandular cells have a round, less heterochromatic nucleus, with a lightly stained cytoplasm and indistinct borders (Figure 3A). In type B glandular cells, the cytoplasmic staining is dark and uniform, with hyperchromatic nuclei and well-defined borders (Figure 3A). *Sp*-SEFIR is mainly located on the membrane of Type B glandular cells (Figure 3B).

### 2.5. Ligand–Receptor Interaction Analysis

Ligand–receptor interaction analysis was conducted using GST pull-down assays. Through prokaryotic expression, we obtained rHisCFSH (20 kDa) (Figure A2), rGSTCFSH (47 kDa) (Figure 4) and rSEFIR (65 kDa) (Figure A3). Results of GST pull-down assay with rGSTCFSH and total protein of AG demonstrated that *Sp*-SEFIR was a CFSH-binding component of AG (Figure 4). The further GST pull-down assay with rHisCFSH and rSEFIR confirmed that *Sp*-CFSH could specifically bind to extracellular regions of *Sp*-SEFIR (Figure 5).

### 2.6. Analysis of Gene Expression in AG following Sp-SEFIR Silencing In Vivo

Previous studies have shown that CFSH could suppress *IAG* and *STAT* expression in *S. paramamosain* [27,28]. To study the involvement of *Sp*-SEFIR in this inhibition, we first carried out in vivo RNA interference experiments. Based on current results, compared to CPS-injected therapy, *Sp*-SEFIR expression was 42% inhibited (Figure 6A). Meanwhile, the knockdown of *Sp*-SEFIR significantly induced *Sp*-IAG and *Sp*-STAT expression in AG (Figure 6B,C).

### 2.7. Analysis of Gene Expression in AG after Medication with rCFSH following Sp-SEFIR Silencing In Vitro

To further explore the involvement of *Sp*-SEFIR in IAG regulation via CFSH, we conducted interference experiments of *Sp*-SEFIR in vitro. According to the results, the addition of *SEFIR*-dsRNA inhibited *Sp*-SEFIR expression by 48% in vitro compared to the PBS therapy (Figure A4A). Furthermore, we confirmed that the addition of rHisCFSH (10^−6^ M) could suppress *Sp*-IAG expression in AG, as previously reported (Figure A4B) [27,28].

Following *Sp*-SEFIR silencing in the in vitro AG explant culture system, we added rHisCFSH (10^−6^ M) and detected the mRNA expression pattern of *Sp*-IAG and *Sp*-STAT. The findings demonstrated that the *knockdown* of *Sp*-SEFIR could also significantly induce *Sp*-IAG and *Sp*-STAT expression in vitro (Figure 7).

## 3. Discussion

In crustaceans, IAG and CFSH are prominent regulators of sexual differentiation [7,14,15,42,43,44,45,46,47]. Although IAG receptors have been identified in many species [8,9,10,11,12,13], no CFSH receptor has been reported until now. Recent studies showed that AG was the principal target organ of CFSH aimed at suppressing IAG expression [15,16,27]. In the mud crab *S. paramamosain*, rCFSH significantly suppressed *IAG* expression in AG explants [27]; a subsequent study by Jiang et al. suggested that CFSH regulated IAG via blocking *STAT* [28]. In the present research, we identified a candidate receptor of CFSH (named *Sp*-SEFIR) involved in the inhibition process via ligand–receptor interaction analysis and biological function studies.

Although a significant number of nucleotide sequences of CFSH have been documented in decapod crustaceans, the IL-17 domain exhibits a high degree of conservation in all identified sequences [29,30]. Interestingly, IL-17 has been identified in various invertebrates, but not in crustaceans [48]. In both vertebrates and invertebrates, IL-17 binds to different IL-17Rs through a functional dimer. Consequently, a signaling system is established between the ligand and the receptor, resulting in the initiation of downstream signals [49,50,51,52]. CFSH might be evolved from an IL-17-like original protein [24], and possibly bound to IL-17R-like polypeptides.

Here, we obtained a transcript encoding an IL-17R-like polypeptide and named it *Sp*-SEFIR. *Sp*-SEFIR shared structures similar to those of other invertebrate IL-17Rs, including a signal peptide, a transmembrane domain, a SEFIR domain and two regions with low compositional complexity. In sea vase *Ciona intestinalis*, the predicted IL-17R protein includes a signal peptide domain at the N-terminal, a transmembrane domain at the central part and a SEFIR domain at the cytoplasmic tail [49]. The classification of *Sp*-SEFIR into IL-17RD was additionally supported by the phylogenetic analysis. The results suggest that *Sp*-SEFIR might be the homologous protein of IL-17RD, and it belongs to the single transmembrane interleukin-17 receptor.

Our results demonstrated that *Sp*-SEFIR had high expression in AG. In a previous study, temporal expression profiles of CFSH and *IAG* during the development of androgenic glands have been identified in male *S. paramamosain* [27]. According to the previous study, CFSH expression showed a significant elevation during stages I and II but a marked decline during stage III. On the contrary, IAG expression levels were relatively lower at stages I–II compared to stage III [27]. In the present study, we noticed that *Sp*-SEFIR and CFSH exhibited similar temporal expression profiles and displayed the opposite expression trend to *IAG*. The current findings suggested that *Sp*-SEFIR may play a role in CFSH’s inhibition of IAG.

On the basis of the SMART software prediction (http://smart.embl-heidelberg.de/ (accessed on 8 December 2022)), *Sp*-SEFIR was a single transmembrane protein. Immunofluorescence analysis further confirmed the membrane localization of *Sp*-SEFIR. To explore the interaction between *Sp-*CFSH and *Sp*-SEFIR, we carried out GST pull-down assays with rHisCFSH and the extracellular segment of *Sp*-SEFIR. The results revealed that the extracellular segment of *Sp*-SEFIR could bind to *Sp-*CFSH. The findings mentioned above indicate that *Sp*-SEFIR is a putative receptor for CFSH. Still, further investigations in appropriate cell lines are needed to verify the putative ligand–receptor interactions between *Sp-*CFSH and *Sp*-SEFIR in the future research.

According to the research, it was proposed that CFSH has the ability to inhibit *STAT* and thereby block *IAG* expression in the mud crab *S. paramamosain* [28]. Both in vivo and in vitro silencing experiments were performed to investigate the probable involvement of *Sp*-SEFIR in this process. We noticed that following *Sp*-SEFIR silencing, both *Sp*-IAG and *Sp*-STAT expression levels significantly increased. Thus, it is reasonable to speculate that *Sp-*CFSH binds to *Sp*-SEFIR, inhibits *Sp*-STAT activity and eventually suppresses *IAG* expression. To verify this hypothesis, in vitro experiments were conducted. The results showed that the addition of rCFSH could suppress *Sp*-IAG expression in AG explants, which was consistent with the previous results [27]. Similar to the result of in vivo experiment, *Sp*-SEFIR silencing could induce *Sp*-IAG and *Sp*-STAT expression in AG explants, indicating that knockdown of *Sp*-SEFIR could relieve inhibition of rCFSH on *Sp*-IAG and *Sp*-STAT expression. Collectively, these results suggest that *Sp*-SEFIR works as a CFSH receptor aimed at regulating *IAG* expression by suppressing *Sp*-STAT expression in *S. paramamosain*.

Furthermore, we noticed that *Sp*-SEFIR exhibited a broad distribution in multiple tissue types, indicating that the biological functions of CFSH are not limited to regulating sexual differentiation. According to the previous study, CFSH probably evolved from an original protein that resembles IL-17 for the conserved IL-17 domain in the mature peptide [14,29,30]. IL-17 is widely involved in the immune response by activating different downstream signal pathways (MAPK, NF-κB and JAK/STAT pathways) in vertebrates and invertebrates [34,37,39,50,51,52,53,54,55]. Based on sequence similarity, we speculate that CFSH likely has an involvement in the immune response of various organs. Further comprehensive investigations are required to elucidate the functions of CFCH in various physiological actions.

## 4. Materials and Methods

### 4.1. Animals

According to previous studies, the development stages of AG were classified as follows: Stage I, AG was small and contained fewer secretory cells, which were attached to the spermaduct; Stage II, AG was clearly linear, cells were clustered or cross-linked into cords, and glands expanded into the surrounding connective tissues; Stage III, AG was largest, and tissue hyperplasia occurred in some areas, which contained the largest number of secretory cells; Stage IV, development of AG stopped, AG degenerated rapidly, and its size was smaller than that in stage II and III [41,56].

In this study, *S. paramamosain* at AG development stage I (body weight: 48.4 ± 4.5 g, carapace width: 6.6 ± 0.5 cm), stage II (body weight: 146.7 ± 9.8 g, carapace width: 11.7 ± 1.0 cm) and stage III (body weight: 249.1 ± 13.2 g, carapace width: 14.7 ± 0.5 cm) were selected as experimental materials. Upon arrival at the laboratory, individuals were subjected to a period of acclimatization, during which they were exposed to controlled conditions consisting of 27 ± 1 °C in addition to a salinity level of 26 ± 0.5 ppm.

### 4.2. cDNA Cloning of Sp-SEFIR

The transcriptome of *S. paramamosain* was utilized to obtain the *Sp*-SEFIR transcript. Moreover, the AG’s total RNA was retrieved by utilizing TRIzol Reagent (Invitrogen, Carlsbad, CA, USA) in accordance with the guidelines provided by the supplier. The 3′ and 5′ untranslated regions (UTR) of *Sp*-CFSHR were acquired through the utilization of rapid amplification of cDNA ends (RACE) technique, employing the SMART ^TM^ RACE cDNA Amplification Kit (Clontech, Palo Alto, CA, USA) in accordance with the producer’s guidelines. The validation of open reading frame (ORF) was performed via synthesizing the first-strand cDNA from 1 μg of total RNA utilizing the PrimeScript RT Reagent Kit with gDNA Eraser (TaKaRa, Dalian, China). The ORF of *Sp*-SEFIR was confirmed by specific primers *Sp*-SEFIR-OF/OR (Table A1). Utilizing LA Taq polymerase (TaKaRa, Dalian, China), polymerase chain reaction (PCR) was conducted as per these criteria: 94 °C for 5 min; 35 cycles of 94 °C for 30 s, 56 °C for 30 s and 72 °C for 100 s, and then through 72 °C for 10 min final extension. PCR outcomes were observed with 1.5% agarose (Biowest, Kansas City, MO, USA) gel electrophoresis before being connected to the pMD19-T vector (Takara, Dalian, China) for sequencing. Table A1 lists the primer sequences.

### 4.3. Quantitative Real-Time PCR (qRT-PCR) Assays

The primers utilized for qRT-PCR were obtained from previous investigations [27,28]. The determination of the amplification effect of all primer pairs was conducted prior to their utilization in qRT-PCR tests. The cDNA underwent a dilution of a four-fold magnitude utilizing water that was free of RNases. Components in a 20 μL qRT-PCR reaction system were: 10 μL of 2 × PCR main mixture containing SYBR Green, 2 μL of diluted cDNA, 0.5 μL of every primer and 7 μL of water. The experiment was conducted using a 7500 rapid RT-PCR (Applied Biosystems, CA, USA), and the experimental parameters utilized for the reaction were: 95 °C for 2 min, then 40 cycles of 95 °C for 15 s, 58 °C for 30 s, and 72 °C for 30 s. The 2^−ΔΔCt^ approach was employed to measure the outcome, with the reference gene being *β-actin* (GenBank accession number: GU992421).

### 4.4. Tissue Distribution of Sp-SEFIR

The reverse transcription-PCR (RT-PCR) was utilized to detect the distribution profile of *Sp*-SEFIR in various tissues (eyestalk ganglion, cerebral ganglion, thoracic ganglion, Y organ, heart, testis, androgenic gland, stomach, hepatopancreas, muscle and epidermis) of *S. paramamosain* (n = 3). The procedures for extracting total RNA and first-strand cDNAs synthesized in accordance with the guidelines outlined in the relevant Section 4.2. The *Sp*-SEFIR-F/R was used as a primer, *β-actin* (GenBank accession no: GU992421) was amplified as a positive control, which was achieved utilizing Ex Taq polymerase (TaKaRa, Dalian, China), and the circumstances were: 94 °C for 5 min; 35 cycles of 94 °C for 30 s, 55 °C for 30 s and 72 °C for 30 s before an extension at 72 °C for 5 min. The PCR outcomes underwent analysis through the utilization of 1.5% agarose gel electrophoresis and were subsequently imaged with a UV detector (Geldoc, Thermo Fisher Scientific, Madrid, Spain).

### 4.5. Sp-SEFIR Expression Profile throughout AG Development

qRT-PCR determined the *Sp*-SEFIR mRNA expression patterns in AG at stage I–III (n = 5). Total RNA was extracted, first-strand cDNAs were synthesized, and qRT-PCR was conducted as mentioned before (Section 4.2 and Section 4.3).

### 4.6. Immunofluorescence Assays

Immunofluorescence assays for *Sp*-SEFIR were performed with AG attached to the subterminal portion of ejaculatory ducts. We entrusted Shanghai GL Biochem Co., Ltd. (Shanghai, China) to produce the *Sp*-SEFIR antibody.

The tissues were first preserved in modified Bouin’s fixative (25 mL 37–40% formaldehyde, 75 mL saturation picric acid in addition to 5 mL glacial acetic acid) for one night at 4 °C. Following gradient alcohol drying, tissues were immersed in paraffin and prepared for 5 μm slices. After being dewaxed and rehydrated, parts of tissue sections were subjected to Hematoxylin-Eosin (HE) staining for histological observation. Meanwhile, other tissue sections were repaired in antigen repair solution (EDTA Antigen Retrieval Solution, pH 8.0; Sangon Biotech, Shanghai, China) at 99 °C for 20 min and cooled down to room temperature naturally. After antigen retrieval, slides were blocked with 5% bovine serum antigen (BSA) in 1 × PBS for 30 min at room temperature. After that, the tissue samples were incubated with *Sp*-SEFIR antibodies (1:500 dilution) or preimmune serum (negative control) at 37 °C for 2 h. Upon using PBS for washing, tissue samples were incubated with Goat Anti-Mouse IgG H&L (Alexa Fluor^®^ 488) preadsorbed (1:100, Abcam, Cambridge, MA, USA) diluted with 1% BSA in 1 × PBS for 45 min at room temperature. Subsequently, tissue samples were incubated with DAPI (1 μg/mL, Beyotime, Shanghai, China) for nuclear staining. Following washing one time in PBS, the slides were locked up using mounting medium, antifading (Solarbio, Beijing, China) and photographed with the BGIMAGING Cellview 4.11 system.

### 4.7. Expression of Sp-CFSH Recombinant Proteins and Sp-SEFIR Extracellular Domain Recombinant Protein

In this study, we expressed two recombinant *Sp*-CFSH proteins: *Sp-*CFSH recombinant protein with only 6 × His tag (rHisCFSH) and *Sp-*CFSH recombinant protein containing both 6 × His tag and GST (rGSTCFSH). rHisCFSH was expressed and purified according to the previous study [27,28]. rGSTCFSH was also expressed utilizing a prokaryotic expression system. The fragment encoding mature peptide of *Sp*-CFSH was cloned into the pET-GST vector, utilizing BamH I and Nhe I restriction enzyme sites. The generated constructs (pET-GST-CFSH-His), which included GST and 6 × His tags, were transformed into *E. coli TransB* (DE3) and induced for 20 h at 16 °C, following the application of isopropyl-beta-D-thiogalactopyranoside (IPTG) at a level of 1 mM. Upon harvest using centrifugation (8000× *g*, 10 min, 4 °C), the bacteria were disrupted by ultrasonic waves. We selected the supernatant for further purification with glutathione sepharose 4B (Solarbio, Beijing, China) according to the guidelines.

The recombinant protein of *Sp*-SEFIR extracellular fragment containing GST and 6 × His tag (rSEFIR) was also expressed with the same prokaryotic expression system as rGSTCFSH. The fragment encoding the extracellular segment of *Sp*-SEFIR was cloned into the pET-GST vector with restriction enzyme sites (EcoR I and Nhe I). The generated constructs (pET-GST-SEFIR-His) were transformed into *E. coli TransB* (DE3) and induced for 6 h at 25 °C after adding IPTG (0.2 mM final concentration). rSEFIR was purified from the supernatant of crude cell extracts with glutathione sepharose 4B (Solarbio, Beijing, China).

### 4.8. GST Pull-Down Assays

Herein, we conducted two GST pull-down assays to detect the protein–protein interaction between *Sp-*CFSH and *Sp*-SEFIR. First, we performed GST pull-down assay with rGSTCFSH and the total protein of AG to detect whether *Sp*-SEFIR was a CFSH-binding component of AG. After that, a GST pull-down assay with rHisCFSH and rSEFIR was performed, aimed at exploring binding regions of *Sp*-SEFIR.

The total protein of AG was extracted with PP11-Universal Protein Extraction Reagent (Aidlab, Beijing, China) according to the instructions. rGSTCFSH was obtained as previously described (Section 4.7). Upon being incubated at 4 °C for 30 min, rGSTCFSH was immobilized in the Glutathione Sepharose 4B (Solarbio, Beijing, China). Then, the total protein of AG was added and incubated at 4 °C for 2 h, and the unbound protein was washed away using PBS. A column volume of glutathione elution buffer was added and incubated for 10 min to elute bound proteins, and the supernatant was collected by centrifugation. The proteins that were eluted underwent analysis through the utilization of SDS-PAGE and Western blot techniques, with the aid of the *Sp*-SEFIR antibody and the anti-His mouse monoclonal antibody. GST protein (with 6 × His tag) was utilized as the negative control.

rHisCFSH and rSEFIR were obtained as described above (Section 4.7). Upon being incubated at 4 °C for 30 min, rSEFIR was immobilized on the Glutathione Sepharose 4B (Solarbio, Beijing, China). Subsequently, rCFSH was introduced and subjected to incubation at a temperature of 4 °C for 1 h. The protein that did not bind was eliminated through PBS washing. A column volume of glutathione elution buffer was added and incubated for 10 min to elute the bound proteins, and the supernatant was collected by centrifugation (500× *g*, 5 min, 4 °C). The eluted proteins were analyzed by SDS-PAGE and Western blot using anti-His mouse monoclonal antibody. GST protein (with 6 × His tag) was used as the negative control.

### 4.9. Silencing Experiment In Vivo

Fragments of *Sp*-SEFIR and green fluorescent protein gene (*GFP*) were cloned into pGEMT-Easy Vector to prepare the linearized DNA templates. The dsRNA synthesis was performed using T7 and SP6 polymerase. *GFP* dsRNA was synthesized as the negative control. Individuals (body weight: 48.4 ± 4.5 g, carapace width: 6.6 ± 0.5 cm, n = 27) in stage I of AG development were equally divided at random into three groups: *GFP*-dsRNA-injected, *SEFIR*-dsRNA-injected and crustacean physiological saline (CPS)-injected [57]. Crabs were injected with either 1 μg/g of dsRNA or an equivalent amount of CPS. The injection was repeated 24 h after the first injection. Then, 72 h after first injection, crabs were anesthetized on ice for 5 min and then AGs were dissected. The interference efficiency of *Sp*-SEFIR and expression levels of *Sp*-IAG as well as *Sp*-STAT, were detected using qRT-PCR. Furthermore, the RNAs extraction, cDNA synthesis and qRT-PCR analysis of samples were conducted in accordance with the methods previously outlined.

### 4.10. In Vitro Experiment: Sp-SEFIR Interference

The in vitro explant culture system was modified and used to investigate the regulatory role of *Sp*-SEFIR in the inhibitory process of *IAG* expression mediated by CFSH [27].

Individuals (body weight: 146.7 ± 9.8 g, carapace width: 11.7 ± 1.0 cm, n = 7) at AG stage II were selected for in vitro experiments. The AG explants of the same individual were split into three equal groups and treated with 200 μL L-15 medium containing *GFP-*dsRNA (1 μg/mL final), *SEFIR-*dsRNA (1 μg/mL final), or CPS with the identical quantity, respectively. Upon incubating at 26 °C for 6 h, the culture media was changed with 200 μL of L-15 media that contained a concentration of 10^−6^ M rCFSH. After a 12 h cultivation period, AG explants were collected to extract total RNA. Expression levels of *Sp*-IAG, *Sp*-STAT were detected using qRT-PCR. RNA extraction, cDNA synthesis, as well as qRT-PCR analysis of samples, were performed according to the criteria mentioned above (Section 4.2 and Section 4.3).

### 4.11. Bioinformatics Analyses

ORF of *Sp*-SEFIR was predicted with the ORF Finder program (https://www.ncbi.nlm.nih.gov/orffinder/ (accessed on 7 December 2022)). The amino acid sequences of *Sp*-SEFIR were subjected to analysis utilizing EXPASY (https://web.expasy.org/protparam/ and http://web.expasy.org/compute_pi/ (accessed on 7 December 2022)). The signal peptide of *Sp*-SEFIR was predicted by SignalP 5.0 Server (https://services.healthtech.dtu.dk/service.php?SignalP-5.0 (accessed on 8 December 2022)). The transmembrane domain and conserved domain were predicted by SMART (http://smart.embl-heidelberg.de/ (accessed on 8 December 2022)) and Phobius (http://www.ebi.ac.uk/Tools/pfa/phobius/ (accessed on 8 December 2022)). IL-17R sequences were from published articles and GenBank library (Table A2). In addition, the construction of a phylogenetic tree was carried out utilizing the neighbor-joining approach (NJ) using MEGA 6.0. The process of bootstrap sampling was iterated 1000 times.

### 4.12. Statistical Analysis

Statistical analysis was conducted utilizing SPSS 18.0 program. The provided data exhibited a normal distribution in accordance with the Kolmogorov–Smirnov test. Levene’s test was employed to assess the homogeneity of variance, and significant variations were determined employing one-way analysis of variance (ANOVA), subsequently employing the Duncan test. *p* < 0.05 was judged significant, while *p* < 0.01 was judged extremely significant. All findings are expressed as mean ± SEM.

## 5. Conclusions

To summarize, we detected a CFSH receptor aimed at regulating IAG expression via the protein interactions experiment and the biological function experiment. As far as we know, this study represents the initial documentation of CFSH receptors. Furthermore, we confirm that CFSH regulates *IAG* expression in AG through the CFSH-SEFIR-STAT-IAG axis in the mud crab *S. paramamosain*. Our findings, as mentioned earlier, offer novel perspectives on molecular pathways that have roles in differentiating sex in crustaceans, in addition to substantiating the pleiotropic effects of CFSH.

## Figures and Tables

**Figure 1 ijms-24-12240-f001:**
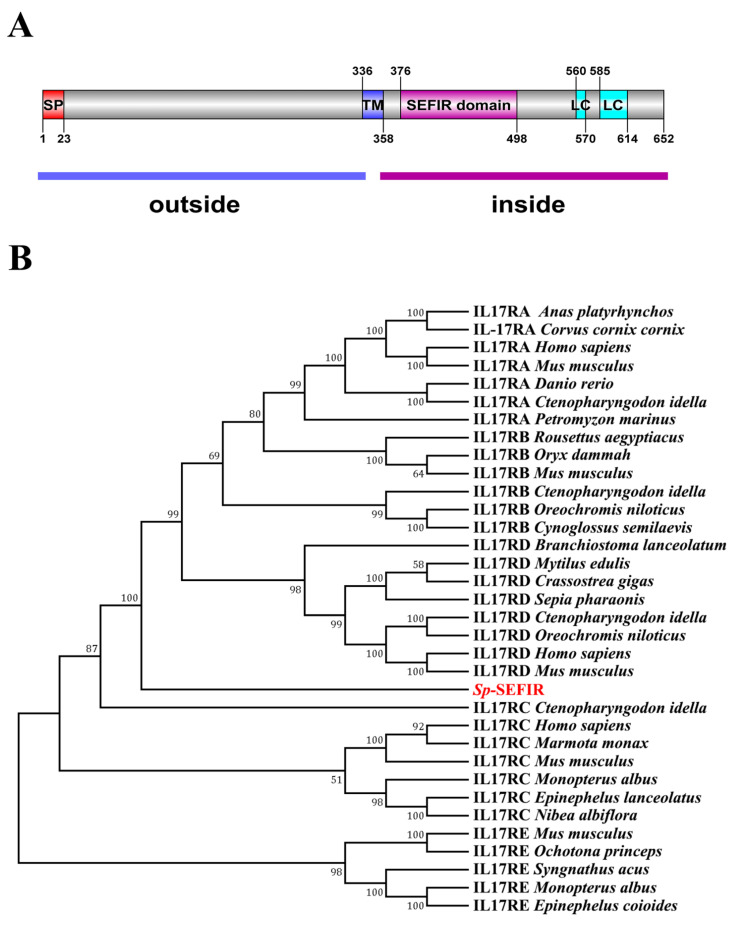
Schematic diagram and phylogenetic tree analysis of *Sp*-SEFIR. (**A**) A schematic diagram was predicted. SP: signal peptide; TM: transmembrane domain; LC: low compositional complexity area. The blue bar marks the extracellular segment; the purple bar marks the intracellular segment. (**B**) The phylogenetic tree was created with conserved SEFIR domain of IL-17R utilizing the NJ approach. The bootstrap test (1000 replicates) was utilized to display the proportion of replicate trees wherein the related taxa clustered together, and this information was presented adjacent to the branches. *Sp*-SEFIR was marked in red.

**Figure 2 ijms-24-12240-f002:**
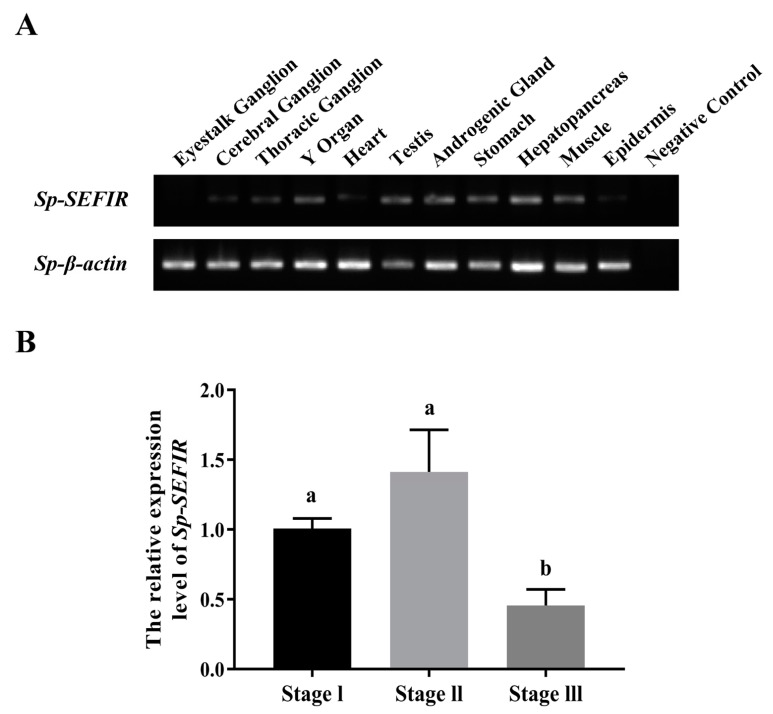
Spatial and temporal expression profiles of *Sp*-SEFIR in male *S. paramamosain*. (**A**) Tissue distribution profile of *Sp*-SEFIR. (**B**) *Sp*-SEFIR expression profile throughout AG development. The standardized expression levels of *Sp*-SEFIR, normalized by *β-actin* expression patterns, were expressed as the mean ± SEM. Statistical analysis was performed utilizing one-way analysis of variance (ANOVA) and then using Duncan’s multiple range tests; “a and b”, *p* < 0.05; n = 5).

**Figure 3 ijms-24-12240-f003:**
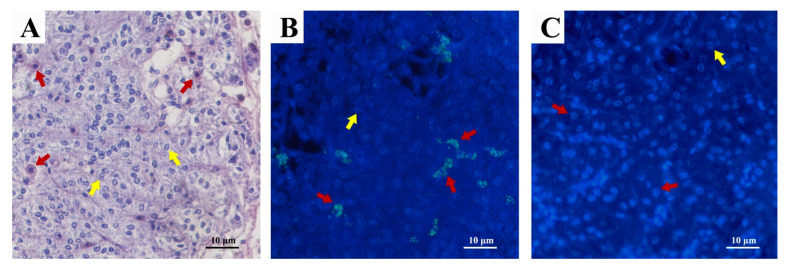
Immunofluorescence localization of *Sp*-SEFIR in AG. (**A**) HE staining of the AG. Two types of glandular cells (type A and type B) were observed. Type A glandular cells have a round nucleus with one or two nucleoli, with a lightly stained cytoplasm and indistinct borders. In type B glandular cells, the cytoplasmic staining is dark and uniform, with hyperchromatic nuclei and well-defined borders. Immunofluorescence localization of *Sp*-SEFIR was performed with mouse anti-*Sp*-SEFIR serum (**B**) or mouse preimmune serum (negative control) (**C**). *Sp*-SEFIR was mainly located on the membrane of Type B glandular cells (**B**). Solid yellow arrows indicated type A glandular cells; solid red arrows indicated type B glandular cells.

**Figure 4 ijms-24-12240-f004:**
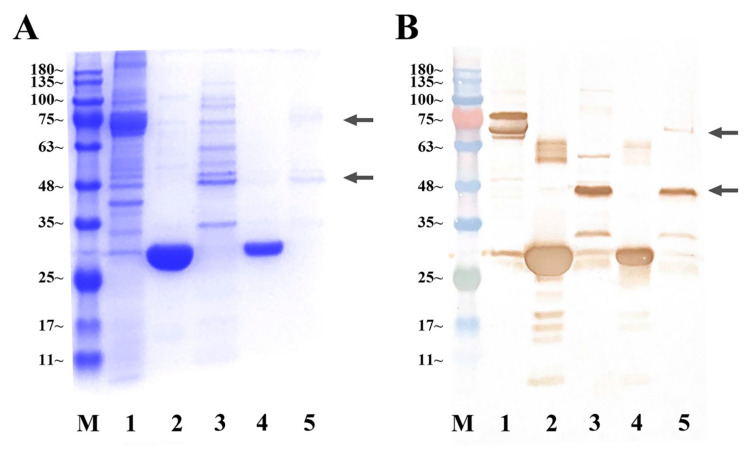
Pull-down assays to detect the interaction between rGSTCFSH and *Sp*-SEFIR. SDS-PAGE analysis (**A**) and Western blot analysis (**B**) of GST pull-down assays. Lane M, protein marker; Lane 1: total protein of AG; Lane 2: initially purified GST protein; Lane 3: initially purified rGSTCFSH; Lane 4: Eluent of beads after co-incubation of GST protein and total protein of AG; Lane 5: Eluent of beads after co-incubation of rGSTCFSH and total protein of AG. rGSTCFSH and *Sp*-SEFIR are marked with arrows. *Sp*-SEFIR, ~74 kDa; rGSTCFSH, ~47 kDa; GST protein, ~28 kDa.

**Figure 5 ijms-24-12240-f005:**
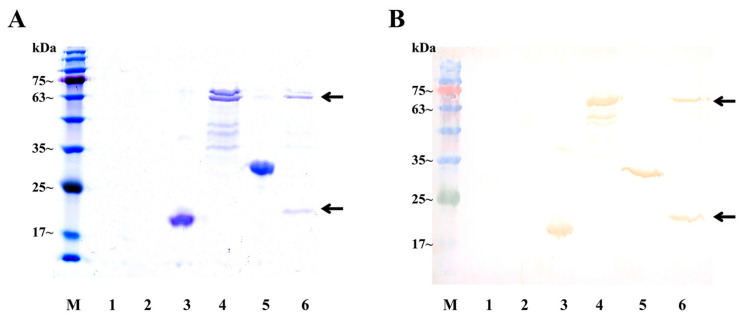
Pull-down assays to detect the interaction between rHisCFSH and rSEFIR. SDS-PAGE analysis (**A**) and Western blot analysis (**B**) of GST pull-down assays. Lane M, protein marker; Lane 1: Fifth PBS washing solution after co-incubation of GST protein (with his tag) and rHisCFSH; Lane 2: Fifth PBS washing solution after co-incubation of rSEFIR and rHisCFSH; Lane 3: rHisCFSH; Lane 4: rSEFIR; Lane 5: Eluent of beads after co-incubation of GST protein and rHisCFSH; Lane 6: Eluent of beads after co-incubation of rSEFIR and rHisCFSH. rHisCFSH and rSEFIR are marked with arrows.

**Figure 6 ijms-24-12240-f006:**
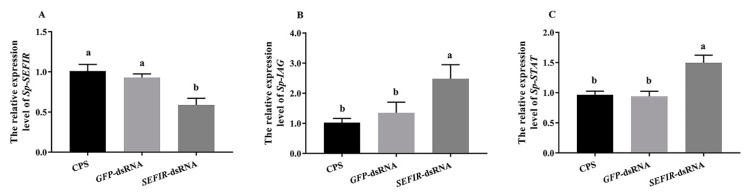
Impacts of short-term *Sp*-SEFIR silencing on gene expression in vivo. Expression levels of *Sp*-SEFIR (**A**), *Sp*-IAG (**B**), and *Sp*-STAT (**C**) were determined after in vivo injection with CPS, *GFP* dsRNA or *SEFIR* dsRNA. The gene expression levels were standardized by *β-actin* expression levels and expressed as mean ± SEM (“a and b”, *p* < 0.05; one-way ANOVA followed by Duncan’s multiple range tests; n = 9).

**Figure 7 ijms-24-12240-f007:**
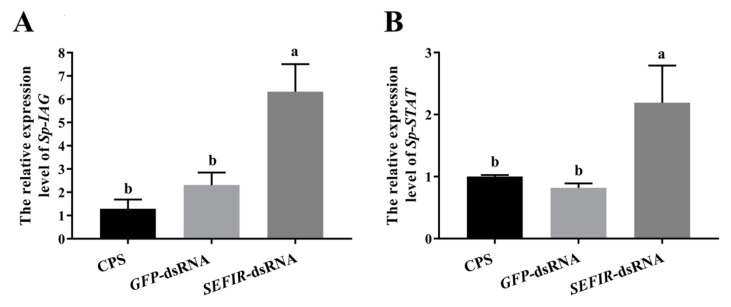
Impacts of rCFSH on gene expression in AG after silencing *Sp*-SEFIR in vitro. Expression levels of *Sp*-IAG (**A**) and *Sp*-STAT (**B**) were determined. The gene expression levels were standardized by *β-actin* expression levels and represented as mean ± SEM (“a and b”, *p* < 0.05; one-way ANOVA; therefore, using Duncan’s multiple range tests; n = 7).

## Data Availability

The data presented in this study are openly available in GenBank [accession number GU992421].

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
