# Peer review of "Identification of a Putative CFSH Receptor Inhibiting IAG Expression in Crabs"

_ijms, 2023, doi:10.3390/ijms241512240_

Round 1
Reviewer 1 Report
Comments about the manuscript:
“Identification of a Putative CFSH Receptor Inhibiting IAG Expression in Crabs”
In crustaceans, the female sex hormone (CFSH) plays a role in sexual differentiation, in particular by inhibiting the expression of the androgen gland factor (IAG). However, the CFSH receptor is still unknown in crustaceans. The study presented here was thus devoted to the demonstration of a receptor (Sp-SEFIR) in Scylla paramamosain whose activity in the regulation of the expression of IAG in the androgen gland has been confirmed.
This well-conducted work is useful for understanding the reproductive biology of a crustacean and could be published after a few minor corrections. Here are some remarks.
Page 2, line 92 “(Figure A1)”: is it not Figure 1A?
Page 5, figure 3. “(A) HE staining of the AG.”: explain how types A and B glandular cells can be distinguished with HE? Specify in the legend.
“(C) Negative control of immunofluorescence assays.”: with the use of pre-immune serum? please specify in the legend.
Page 5, line 136. “(Figure A2)”: is it not Figure 2A?
Page 5, line 137. “(Figure A3)”: is it not Figure 3A?
Page 7, lines 175, 176: is it not “Figure 4A” and “Figure 4B” instead of “A4A” and “A4B”? Please, check.
Page 11, lines 339-340. “the harvest using centrifugation”: Give the characteristics of centrifugation: speed in g number, time?
Author Response
Thank you for your constructive comments. We are appreciated to have a chance to improve our manuscript. Below we address your comments in a point-to-point response.
Point 1: Page 2, line 92 “(Figure A1)”: is it not Figure 1A?.
Response 1: Thanks for your suggestions. This refers to the Appendix Figure A1 which is labeled as Figure A1 according to author guidelines.
Point 2: Page 5, figure 3. “(A) HE staining of the AG.”: explain how types A and B glandular cells can be distinguished with HE? Specify in the legend.
Response 2: Thanks for your suggestions. Type A glandular cells have a round nucleus with one or two nucleoli, with a lightly stained cytoplasm and indistinct borders; In type B glandular cells, the cytoplasmic staining was dark and uniform, with hyperchromatic nuclei and well-defined borders. We have been made modification accordingly. Please see Legend of Figure 3, line 134-138.
Point 3: “(C) Negative control of immunofluorescence assays.”: with the use of pre-immune serum? please specify in the legend.
Response 3: Thanks for your suggestions. Changes have been made modification accordingly. Please Legend of Figure 3, line 138-140.
Point 4: Page 5, line 136. “(Figure A2)”: is it not Figure 2A?
Response 4: Thanks for your suggestions. This refers to the Appendix Figure A2 which is labeled as Figure A2 according to author guidelines.
Point 5: Page 5, line 137. “(Figure A3)”: is it not Figure 3A?
Response 5: Thanks for your suggestions. This refers to the Appendix Figure A3 which is labeled as Figure A3 according to author guidelines.
Point 6: Page 7, lines 175, 176: is it not “Figure 4A” and “Figure 4B” instead of “A4A” and “A4B”? Please, check.
Response 6: Thanks for your suggestions. This refers to the Appendix Figure A4 which is labeled as Figure A4 according to author guidelines.
Point 7: Page 11, lines 339-340. “the harvest using centrifugation”: Give the characteristics of centrifugation: speed in g number, time?
Response 7: Thanks for your suggestions. Bacterial bodies were pelleted by centrifuging at 4 °C, 8000×g for 10 min. Modifications have also been made accordingly. Please see line 352.

Reviewer 2 Report
The paper by Liu et al. presents the molecular characterization of the cognate receptor for CFSH, Sp-SEFIR, from the mud crab Scylla paramamosain. The authors demonstrated cDNA cloning and tissue distribution of Sp-SEFIR, interaction of Sp-SEFIR with CFSH using a pull-down binding assay, and RNAi-based evaluation of Sp-SEFIR. The methods are appropriate, and the results are sufficient and convincing. Please consider to improve the following minor points.
(1) The molecular phylogenetic tree (Fig. 1B) includes too low bootstraps such as "6", "2", "11".... This result indicates that the phylogenetic tree is not statistically significant and not of scientific value. Hence, the authors should provide an improved phylogenetic tree with all bootstraps higher than 70 focusing on highly conserved regions of IL17R and/or putative SEFIR in other animals. Otherwise, it is an idea that the authors remove the phylogenetic tree and, instead, provide sequence alignments of Sp-SEFIR with L17Rs and/or putative SEFIRs.
(2) The direct evidence for binding of CFSH to Sp-SEFIR expressed inappropriate cultured cells are required in the future research. The authors should state it in the Discussion section.
(3) Overall, the paper has been well-written and understandable, but several tiny errors were found:
Line 196. Replace "nuclear" with "nucleotide".
Line 204. "protein" or "polypeptide" is better than "peptide".
Line 227. "putative" is better than "probable".
Therefore, the authors recommends that the manuscript is subjected to an English editing service.
The manuscript may be subjected to an English editing service.
Author Response
Thank you for your constructive comments. We are appreciated to have a chance to improve our manuscript. Below we address your comments in a point-to-point response.
Point 1: The molecular phylogenetic tree (Fig. 1B) includes too low bootstraps such as "6", "2", "11".... This result indicates that the phylogenetic tree is not statistically significant and not of scientific value. Hence, the authors should provide an improved phylogenetic tree with all bootstraps higher than 70 focusing on highly conserved regions of IL17R and/or putative SEFIR in other animals. Otherwise, it is an idea that the authors remove the phylogenetic tree and, instead, provide sequence alignments of Sp-SEFIR with L17Rs and/or putative SEFIRs.
Response 1: We are sorry for the unpersuasive molecular phylogenetic tree. Herein, we reconstructed phylogenetic tree with conserved regions of IL17R. We replaced some sequences as some genomic sequences record were removed because they have been superseded by a new assembly of the genome. Please see Figure 1B, Table A1 and line 104.
Point 2: The direct evidence for binding of CFSH to Sp-SEFIR expressed in appropriate cultured cells are required in the future research. The authors should state it in the Discussion section.
Response 2: Thanks for your suggestions. We have added corresponding texts in Section 3 Discussion. Please see line 237-239.
Point 3: Overall, the paper has been well-written and understandable, but several tiny errors were found:
Line 196. Replace "nuclear" with "nucleotide".(199)
Line 204. "protein" or "polypeptide" is better than "peptide".(207)
Line 227. "putative" is better than "probable".(230)
Therefore, the authors recommends that the manuscript is subjected to an English editing service.
Response 3: Thanks for point out our mistakes. We have made made modifications to the manuscript accordingly. Please see line 205, 212 and 236. Meanwhile, the revised manuscript has been polished by an English language editing company.
